# *TP53* in Myelodysplastic Syndromes: Recent Biological and Clinical Findings

**DOI:** 10.3390/ijms21103432

**Published:** 2020-05-13

**Authors:** Cosimo Cumbo, Giuseppina Tota, Luisa Anelli, Antonella Zagaria, Giorgina Specchia, Francesco Albano

**Affiliations:** Department of Emergency and Organ Transplantation (D.E.T.O.), Hematology Section, University of Bari, 70124 Bari, Italy; cosimo.cumbo@gmail.com (C.C.); giuseppina.tota@uniba.it (G.T.); luisa.anelli@uniba.it (L.A.); antonellazagaria@hotmail.com (A.Z.); specchiagiorgina@gmail.com (G.S.)

**Keywords:** *TP53* mutation, p53 expression, myelodysplastic syndrome, del(5q), prognosis, target therapy

## Abstract

*TP53* dysregulation plays a pivotal role in the molecular pathogenesis of myelodysplastic syndromes (MDS), identifying a subgroup of patients with peculiar features. In this review we report the recent biological and clinical findings of *TP53*-mutated MDS, focusing on the molecular pathways activation and on its impact on the cellular physiology. In MDS, *TP53* mutational status is deeply associated with del(5q) syndrome and its dysregulation impacts on cell cycle, DNA repair and apoptosis inducing chromosomal instability and the clonal evolution of disease. *TP53* defects influence adversely the MDS clinical outcome and the treatment response rate, thus new therapeutic approaches are being developed for these patients. *TP53* allelic state characterization and the mutational burden evaluation can therefore predict prognosis and identify the subgroup of patients eligible for targeted therapy. For these reasons, in the era of precision medicine, the MDS diagnostic workup cannot do without the complete assessment of *TP53* mutational profile.

## 1. Introduction

Myelodysplastic syndromes (MDS) are a group of clonal hematopoietic stem cell (HSC) malignancies characterized by bone marrow dysplasia, ineffective hematopoiesis leading to peripheral blood cytopenia, and by the risk of acute myeloid leukemia (AML) transformation [1]. MDS are a group of diseases with a high degree of variability in terms of prognosis, clinical phenotype and response to treatment. This heterogeneity can often be associated to a high genotypic variability among affected individuals, highlighted in the past decade owing to the application of new high throughput technologies, including microarray analysis and next-generation sequencing (NGS) [2,3]. Large-scale analysis of the molecular mechanisms of the disease has enabled the identification of a set of genes that are recurrently mutated in MDS. They are involved in different cellular processes, such as histone modification (e.g., *ASXL1*, *EZH2*) and DNA methylation (e.g., *TET2*, *DNMT3A*, *IDH1*, *IDH2*), signal transduction (e.g., *NRAS*, *JAK2*), transcriptional regulation (e.g., *RUNX1*, *TP53*), and RNA splicing (e.g., *SF3B1*, *SRSF2*, *U2AF1*, *ZRSR2*) [4,5]. In this variety of genes, the *Homo sapiens tumor protein p53* (*TP53*) dysregulation plays a crucial role in MDS phenotype, treatment response, and risk of AML transformation [6,7].

*TP53* is a tumor suppressor gene that spans 19,144 bp on chromosome 17p13.1 and contains 11 exons. The protein has five functional domains: The transactivation domain and a proline-rich domain in the N-terminal region; the oligomerization domain and a regulatory domain in the C-terminal region; the DNA-binding domain (DBD) in the central core [8,9]. The protein is an essential transcription factor for cell cycle arrest, DNA repair mechanisms, apoptosis induction, and cellular differentiation regulation [10,11]. *TP53* plays a pivotal role in the cellular apoptotic response to DNA damaging agents, such as cytotoxic chemotherapy and its dysregulation is generally associated with a negative prognostic impact in oncologic diseases [12,13]. *TP53* is the gene most closely studied in cancer, and its role is widely documented in different hematological malignancies: in lymphoid neoplasms such as chronic lymphocytic leukemia (CLL) and acute lymphoblastic leukemia (ALL) and in myeloid diseases such as AML [14]. 

Herein we address *TP53*-mutated MDS, summarizing the recent biological and clinical findings in this patients subgroup. A first section is reserved to the molecular aspects of *TP53* dysregulation: acquired or constitutive mutations and protein expression, with a special focus on cellular pathways activation and on *TP53* correlations with karyotype aberrations. The prognostic value of *TP53* and its influence on treatment decision-making is also discussed, considering the emerging therapeutic strategies that are currently being developed.

## 2. Biological and Molecular Aspects

### 2.1. Molecular Pathways Activation

*TP53* is the most commonly mutated gene in human cancer. Its mutational state in MDS is strongly associated with solitary del(5q) (~20%), or complex karyotypes (CK) with -5/5q- (~70%) [15,16]. For this reason, the majority of studies has explored the association of p53 to del(5q) MDS. Deletion of the long arm of chromosome 5 causes the loss of 1.5 megabases, the “commonly deleted region” (CDR), comprising 41 genes situated close to or within 5q32-33 [17,18,19]. Among all the 41 genes in the CDR, those that may play pivotal roles in tumorigenesis include: *RSP14,* which is important in ribosomal function and RNA synthesis, *miR145* and *miR146,* that intervene in innate immunity and signaling, *CDC25c/PP2A,* a phosphatase that regulates cell division, *SPARC,* that mediates adhesion and *EGR1* and *DIAPH,* which act as tumor suppressor and cytoskeleton organizer, respectively [18,19,20]. Only with *RPS14* gene suppression were the maturation and proliferation of erythroid precursors halted, reproducing the del(5q) syndrome phenotype [18]. Moreover, *RPS14* haploinsufficiency was correlated to an enhanced p53 expression in an in vivo model, together with age-dependent progressive anemia, dysmegakaryopoiesis, modification of the stem cell niche, and loss of hematopoietic stem cell quiescence [21]. Additional studies demonstrated that after blocking Murine Double Minute-2 (MDM2) using the small molecule Nutlin, p53 was stabilized and activated, a condition that compromised erythropoiesis in a similar way to del(5q) MDS [22,23]. In normal conditions, MDM2 is free to bind p53 and MDM2-p53 binding determines p53 ubiquitination and consequent degradation, in a normal cell cycle (Figure 1A). *RPS14* haploinsufficiency in del(5q) MDS triggers ribosomopathies typified by nucleolar stress, in which ribosome assembly is impeded and small ribosomal proteins (RPs) do not bind to 40S and 60S ribosomal subunits, but are free to bind to MDM2. MDM2-RPs binding prevents MDM2-p53 interaction, resulting in p53 stabilization. This abnormal accumulation of p53 leads to cell cycle arrest, impaired DNA repair, senescence, and apoptosis (Figure 1B). Apoptosis in maturing erythroids occurs at the step converting polychromatic to orthochromatic erythroblasts, provoking erythroid hypoplasia, a typical feature of del(5q) MDS [21]. Moreover, cytotoxic stresses activate the phosphorylation of both MDM2 and p53 by ATM-Chk1 or ATM-Chk2, thus activating other post-translational modifications, including acetylation, methylation, or sumoylation of MDM2, which reinforce p53 activity [24,25] (Figure 1B). In this altered pathway, lenalidomide treatment can intervene, although *TP53* mutations in del(5q) MDS alter treatment responses and increase the risk of leukemia transformation [26]. 

Among its various mechanisms, lenalidomide induces proteasome degradation of CK1a, the protein product of *CSNK1A1*, one of the CDR genes [27,28]. Through its interactions with MDM2 and with the b-catenin destruction complex, CK1a negatively regulates both p53 and b-catenin protein levels. In del(5q) MDS, a haploinsufficient expression of CK1a plays an important role in the disease pathogenesis and treatment response rate. In fact, lenalidomide induces the degradation of CK1a, which can be tolerated by normal cells with two copies of *CSNK1A1*, but results in p53-mediated apoptosis of del(5q) cells because of their haploinsufficient expression of CK1a [27]. Furthermore, the overexpression of *CSNK1A1* reduces the lenalidomide sensitivity of BM cells in patients with del (5q) MDS [27].

Few studies have been focused on the role of p53 in non del(5q) MDS. In 2015, two studies explored the influence of single-nucleotide polymorphisms (SNP), in particular on *TP53* R72P and *MDM2* SNP309. The *TP53* R72P mutation features an amino acid variant on the tertiary structure of the p53 binding domain, causing functional variations. The R/R homozygous genotype, because of having a “C” allele, has a better apoptosis-promoting potential than the P/P genotype, in part because of its major mitochondrial positioning, activating cytosolic release of cytochrome C. Instead, the homozygous P/P genotype shows a higher transcriptional efficacy than the R/R genotype, inducing the cell to halt the cell cycle at the G1 level [29]. Moreover, the *MDM2* SNP309 “G” allele enhances *MDM2* expression and consequently decreases p53 expression. When predicting p53 activity (based on TP53 R72P and MDM2 SNP309), it was demonstrated that non-del(5q) MDS patients with a high p53 SNP activity had a significantly longer overall survival (OS) and progression-free survival (PFS) compared to patients with a low p53 SNP activity [6,29,30] (Figure 1C). It is noteworthy that in chronic myeloid leukemia, the same mechanism could be the basis of the therapeutic resistance shown by patients treated with different lines of tyrosine kinase inhibitors treatment [31,32].

Although it is difficult to identify how p53 deregulations in MDS modify pathways within cancer cells, other studies have attempted to correlate the role of p53 in MDS to chromosomal instability and chromothripsis. Chromosomal instability is linked to MDS and it has been hypothesized that MDS arise primarily from DNA repair defects. The incidence of MDS, in fact, is considerably higher in older people and in patients with genetic defects in DNA repair, like Fanconi anemia and Bloom, Werner and Rothmund-Thomson syndromes [33]. Furthermore, almost 50% of patients with MDS exhibit genetic rearrangements and there is evidence that chemo- and radiation therapies dramatically augment the risk for MDS [33]. At first, a xenograft model was used to try to understand the MDS generation of chromosomal instability [34]. Unfortunately, in vitro and xenotransplanted models have a limited ability to reproduce the natural cancer microenvironment [35], thus, this work did not clarify how inactivation of *TP53* directly increases chromosomal instability, inducing CK in MDS, nor how it allows chromosomally unstable HSC to bypass senescence or apoptosis and to survive [34].

However, genetic instability modifies the DNA damage responses (DDR) that are presumably implicated in the pathogenesis of MDS [36]. An increase of double-strand breaks (DSB) was observed in MDS, together with an impaired DDR, due to an altered pattern of phosphorylated DDR key proteins, including *TP53*. A reduced expression of *TP53* after phosphorylation presumably indicated impaired downstream signaling of DNA damage and/or defects in downstream components of the DDR [36].

Chromothripsis, instead, is a genetic aberration in which tens to hundreds of clustered genomic rearrangements occur in a one-step catastrophic event [37]. The molecular foundation of this genomic chaos has long been studied, and several possible mechanisms suggested [37,38]. One of these is somatically acquired *TP53* mutations. In fact, approximately 50% of AML patients carrying *TP53* mutations displayed chromothripsis, and almost all medulloblastomas showing chromothripsis had *TP53* mutations [38]. Most of these mutations were mapped on exon 5 of *TP53*, and all of them were localized in the DBD, which plays a pivotal role in transcriptional transactivation [39]. Chromothripsis was analyzed in 301 MDS samples, carrying out genome-wide analysis of DNA copy number abnormalities and mutational analysis by NGS [39]. Cryptic genomic abnormalities were found in 23.6% of cases, detected mainly in patients with a normal (45%) or non-informative (15%) karyotype by conventional cytogenetics. *TP53* deletion and mutation (15%) was an exception, being identified in patients with a complex karyotype. Three (1.2%) high-risk MDS cases (two RAEB-1, with 6% and 8% of BM blasts, respectively, and one RAEB-2, with 12% of BM blasts) displayed chromothripsis and all carried *TP53* mutations, underlining the association between *TP53* gene alteration and chromosomal abnormalities, and chromothripsis [39] (Figure 1D). 

Moreover, *TP53* was also investigated for its role during MDS disease progression to AML, that occurs in about 30% of patients [40]. Both linear and branching patterns of evolution have been described: In linear evolution, serial dominant clones appear after the acquirement of supplementary mutations, overgrowing their ancestral clone; branching evolution is, instead, characterized by the appearance of various subclones from one common ancestral clone, and by the coexistence of kindred (sub)clones that carry a partially overlapping set of mutations [40,41]. Furthermore, a “clone sweeping” pattern was proposed, in which a new or a pre-existing subclone sweeps out the other clones and predominates throughout the progression. In this pathway, different kinds of mutations decide the progression fate: “type 1 mutations” cause the transformation from MDS to AML, and “type 2 mutations,” that include *TP53*, favor the progression from lower-risk to higher-risk MDS [42]. Starting with the initial mutations, subsequent hits are not random, but occur in a specific order and some mutation conjunctions occur more frequently [43]. For example, *TP53* dominant mutations were more likely to precede secondary *TP53* mutations, but less likely to precede secondary *ASXL1* mutations [43]. Indeed, several *TP53* independent clones may coexist, highlighting a broad genetic intratumoral heterogeneity of human tumors. It was observed that all oncogenic *TP53* variants were localized in alternative alleles, and each *TP53* variant was presumed to belong to an autonomous subclone arising from a wild-type (WT) *TP53* founder clone, that requires *TP53* inactivation for further progression. All of these subclones present a great dynamic evolution, but it is not known whether this evolution is triggered by treatment, by an intrinsic property of the tumor cells or both [44] (Figure 1D). 

More recently, it has been observed that the loss of heterozygosity (LOH) is a frequent but not imperative step during clonal evolution of tumors with *TP53* missense mutations [45]. Normally, a tumor suppressor gene requires a biallelic inactivation, and this unusual *TP53* behavior has led to the formulation of two hypotheses: the first contemplates an oncogenic gain of function (GOF) of *TP53* missense mutants [46], the second envisages a dominant negative effect (DNE), that leads to a selection for *TP53* missense mutations, with a non-mutational impairment of the remaining WT allele [47]. To verify the two hypotheses, isogenic human leukemia cell lines of the most common *TP53* missense mutations were generated, employing CRISPR-Cas9 [48]. Functional, DNA-binding, and transcriptional analyses highlighted loss of function but no GOF effects. Instead, missense variants in the DBD exert a dominant negative effect, demonstrated through comprehensive mutational scanning of p53 single-amino acid variants and also in mice, where the DNE of p53 missense variants bestows a selective DNA damage benefit to hematopoietic cells [48]. It has also been suggested that p53 GOF is not an autonomous cell phenomenon [35]. The p53 mutant protein establishes a new relationship with the microenvironment, which is in turn conditioned by other extrinsic factors. Thus, a new regulatory circuit is constituted between cancer cells and the microenvironment, in which p53 mutants lie at the molecular heart, and are crucial for the outcome. Solid in vivo evidence revealed the existence of p53 GOF but no in vitro evidence exists because of the impossibility of reproducing the tumor microenvironment. Furthermore, p53 mutant proteins respond to extrinsic signals, becoming stabilized. Thus, a convergence of mechanisms retained from the WT protein, and newly gained by the mutant protein, activate the p53 GOF mutant, and the GOF effects might depend on the degree of activation of the p53 mutant protein [35] (Figure 1D). 

A p53 mutant GOF was revealed in clonal hematopoiesis of indeterminate potential (CHIP) [49]. CHIP appears when a single mutant hematopoietic stem and progenitor cell (HSPC) contributes to a considerable clonal amount of mature blood lineages. This condition is common in aged healthy individuals but it is linked to an increased risk of hematological neoplasms, such as MDS and AML [49,50]. A diagnosis of CHIP requires the presence of a somatic mutation with a mutant allele fraction of at least 2% in the peripheral blood and no other evidence of a hematological malignancy [51,52]. *TP53* gene ranks in the top five among genes mutated in CHIP and its mutational state in CHIP is similar to that of hematological malignancies. Indeed, approximately 90% of *TP53* somatic mutations in CHIP are missense variants in the DBD of the p53 protein. It was shown that mutant p53, but not WT protein, engages EZH2, a key component of Polycomb repressive complex 2 (PRC2), conferring a competitive advantage to HSPCs. In fact, the p53-EZH2 bond reinforces the EZH2 linkage with the chromatin, catalyzing the trimethylation of lysine 27 of histone H3 (H3K27me3) in genes regulating HSPC self-renewal and differentiation [49].

Lastly, in MDS, a *TP53* dysregulation resulting from a position effect, as described for other genes in several myeloid neoplasms [53], has never been proved. In such a scenario, *TP53* activity, and its pathological dysregulation, play a pivotal role in the molecular pathogenesis of MDS. As already mentioned, *TP53* pathway alterations have an impact not only on the biology of the cell but also on the clinical onset and the evolution of the disease, identifying a subgroup of patients with similar features. These issues will be developed below.

### 2.2. TP53 Allelic State and Mutational Burden

The majority of *TP53* mutations are missense variants clustering within the DBD. Consistent with its role as a tumor suppressor, bi-allelic targeting is very frequent. In fact, more than 91% of *TP53*-mutant cancers exhibit second allele loss due to mutation, chromosomal deletion (involving 17p13 locus), or copy-neutral LOH (cn-LOH) [54]. The *TP53* allelic state was recently studied in a cohort of 3324 MDS patients, at diagnosis and treatment naïve; 486 mutations were identified across 378 individuals [55]. Four main *TP53* mutational profiles were identified: 1. Mono-allelic mutation (*n* = 125, 33% of *TP53*-mutated patients); 2. multiple mutations without deletion or cnLOH affecting the *TP53* locus (*n* = 90, 24%); 3. mutation(s) and concomitant deletion (*n* = 85, 22%); 4. mutation(s) and concomitant cnLOH (*n* = 78, 21%). Additionally, in 24 (0.7%) patients, the *TP53* locus was affected by deletion (*n* = 12), cnLOH (*n* = 2) or isochromosome 17q rearrangement (*n* = 10), with no evidence of *TP53* mutations. The study shows that the majority (67%) of *TP53*-mutated MDS patients presents a multiple hit consistent with bi-allelic targeting. The authors demonstrated that, even if *TP53* is universally considered as an adverse prognostic biomarker, only the multi-hit *TP53* state in MDS is associated with genome instability and the worst clinical outcome, not the bare presence of any *TP53* mutation [55]. For these reasons, the MDS diagnostic workup needs to be implemented with a more accurate characterization of the *TP53* allelic state. For this purpose, karyotype analysis, still performed by conventional and molecular cytogenetics, must be coupled with NGS approaches, finalized to study the copy-number status of the *TP53* gene, its mutational profile, and the variant allele frequency (VAF) of the mutations identified. Moreover, it has been widely demonstrated that the *TP53* mutational burden can affect prognosis in MDS [56,57,58]. A recent study shows the impact of *TP53* VAF on the MDS phenotype and outcomes in 219 patients with MDS [56]. MDS patients with a VAF > 40% had a median OS of 124 days; the same OS was not reached in patients with VAF < 20% (*p* < 0.01), as validated in an independent cohort (*p* = 0.01). *TP53* VAF further stratified distinct prognostic groups independently of clinical prognostic scoring systems [56]. The *TP53* mutational burden was later studied in a cohort of 154 lower-risk MDS patients, again demonstrating its role in clinical outcome [57]. In fact, evaluation of the OS determined a 6% *TP53* VAF threshold as an optimal cut-off for patient stratification. At diagnosis, the median OS was 43.5 months in MDS patients with a VAF > 6% compared to 138 months in WT patients (*p* = 0.003); similarly, the median PFS was 20.2 months versus 116.6 months (*p* < 0.0001). In contrast, no significant impact on PFS or OS was observed in MDS patients with a VAF < 6%, who remained stable for long periods without progression [57]. The importance of *TP53* VAF was recently demonstrated in a study of 80 MDS patients (and 112 AML patients) who underwent allogeneic hematopoietic stem cell transplantation (allo-HSCT) [58]. In fact, *TP53* and *EZH2* mutations with a VAF > 33% were associated with poor relapse-free survival (RFS) [58]. In view of these considerations, assessment of the mutational status of the *TP53* gene cannot be limited to the simple presence/absence of any mutation, but requires a more complex evaluation. *TP53* allelic state characterization and the mutational burden evaluation must therefore be considered as part of the MDS diagnostic workflow. In view of these considerations, assessment of the mutational status of the *TP53* gene cannot be limited to the simple presence/absence of any mutation, but requires a more complex evaluation. *TP53* allelic state characterization and the mutational burden evaluation must therefore be considered as part of the MDS diagnostic workflow. To this end, great benefit will be offered by the implementation of NGS technologies. Long-read third generation sequencing, already tested in the hematological field [59] and applied for *TP53* mutational analysis in other diseases [60,61,62], could allow a more rapid and better phasing of *TP53* mutations, discriminating between mutations occurring in the same allele (in *cis*) or in different alleles (in *trans*), as already demonstrated for other targeted genes [62,63,64]. However, the accuracy improvement of these emerging technologies will be needed before they will be of clinical utility. 

### 2.3. TP53 Germline Mutations and Familial MDS Predisposition

MDS has conventionally been considered a disease related to aging, with a median age of onset in the 7th decade, consistent with the frequency of age-related clonal hematopoiesis [65,66]. On the contrary, these malignancies in children and young adults are very uncommon; in fact, the annual rate of MDS is 0.2 per 100,000 for patients under 40 years of age but a ~300-fold higher incidence of 58 per 100,000 has been registered in patients over 80 years [67]. Several differences between the phenotype of MDS occurring in young patients and the elderly suggest distinct leukemogenic genetic drivers [68]. Indeed, as long ago as 1990, when Li-Fraumeni syndrome (LFS) was linked to *TP53* germline mutations, the first evidence about the genetic basis of familial leukemia began to emerge [69]. Since this discovery, many other genes (*ANKRD26*, *CEBPA*, *DDX41*, *ETV6*, *GATA2*, *RUNX1*, *SRP72*) associated with a hereditary predisposition to MDS/AML have been described [70]. LFS (OMIM #151623) is an autosomal dominant familial cancer predisposition syndrome caused by *TP53* germline mutations [71,72], with a risk of malignant transformation to MDS and AML estimated at 8% [70]. In LFS individuals, the constitutive *TP53* defect can impact critically on blood cell development and contribute to the emergence of abnormal hematopoietic clones [73]. Genetic screening of 110 MDS samples collected between 1990–2012 was recently performed to investigate the presence of germlines mutations in bone marrow failure genes (*FANCA*, *GATA2*, *MPL*, *RTEL1*, *RUNX1*, *SBDS*, *TERT*, *TINF2*, and *TP53*). Pathological mutations were identified in 15 of 110 (13.6%) pediatric and young adult patients with MDS, three of which were *TP53* constitutional mutations associated to LFS individuals [74]. In 2017, pediatric hematologists-oncologists, geneticists, and genetic counselors, following the Childhood Cancer Predisposition Workshop of the American Association for Cancer Research, released recommendations for the surveillance of individuals with a hereditary predisposition to leukemia. In these patients, tumor surveillance is finalized to early disease detection to allow a prompt initiation of treatment, with the aim of minimizing morbidity and mortality. This is particularly useful for more indolent diseases, such as MDS, that evolve over months to years. In these cases it is often possible to detect progressive cytopenia, bone marrow dysplasia, and the emergence of somatic genetic and cytogenetic abnormalities [73]. In this perspective, centers already performing tumor-only target sequencing for the diagnostic workup of MDS/AML patients are optimizing these tests, to detect germline variants of genes associated with familial forms of MDS/AML (e.g., *RUNX1*, *CEBPA*, *GATA2, TP53*), to maximize detection and reorganize patients management [75].

### 2.4. p53 Protein Expression

As widely discussed, *TP53* mutational analysis by NGS is a fundamental step in the MDS diagnostic workflow, but the approach remains expensive, time-consuming, and limited to few referral centers. On the other hand, immunohistochemistry (IHC) is a fast, reproducible, and cost-effective technology that can be used in every routine laboratory to estimate p53 protein expression in bone marrow core biopsy [76]. In fact, IHC cannot normally detect the WT p53 protein, because of its short half-life. On the contrary, mutated proteins can usually be easily detected in formalin-fixed, paraffin-embedded tissues, because they can accumulate in the nucleus due to their prolonged half-life [77]. Therefore, the IHC detection of p53 protein suggests an underlying mutation in the gene [78]. Furthermore, aberrant expression of the p53 protein was correlated with hemizygous *TP53* deletion in lymphoproliferative diseases such as CLL (*p* < 0.001) and multiple myeloma (MM) (*p* < 0.001) [79,80,81]. In MDS, a strong correlation between p53 overexpression and gene mutations (*p* < 0.05) [15] has been widely documented [15,76,82,83,84]. Importantly, a recent study also showed the association between p53 expression and the *TP53* mutations VAF (*r* = 0.867, *p* < 0.001), bone marrow (BM) blast percentage (*r* = 0.362, *p* = 0.007), cytogenetic characteristics: 17p abnormalities (*p* = 0.012), 17p deletion (*p* = 0.014), 5q deletion (*p* < 0.001), CK (*p* < 0.001), and a worse outcome [76]. The prognostic impact of p53 expression in MDS was recently studied [84,85,86]; its overexpression is associated with a more aggressive clinical outcome and adverse histological prognostic factors, such as BM fibrosis [86,87]. In fact, the degree of BM fibrosis was related to parameters of erythropoietic failure, marrow cellularity, p53 protein accumulation, *WT1* gene expression, and serum levels of *CXCL9* and *CXCL10* [87]. The correlation between BM fibrosis and p53 overexpression supports the hypothesis that patients with BM fibrosis at diagnosis can have a worse clinical outcome [88,89]. In view of these considerations, because sequencing technologies are not always available for *TP53* mutational status characterization, p53 ICH should be considered a feasible alternative to *TP53* sequencing [76].

### 2.5. TP53 and Karyotype Aberrations

Karyotype is one of the main components of the International Prognostic Scoring System (IPSS) and revised IPSS (IPSS-R), that dictate the basis for MDS prognostication. Del(5q), −7/del(7q), del(20q), +8, and −Y are the cytogenetic anomalies most abundantly explored in MDS [90]. Deletion of the long arm of chromosome 5 is the unique cytogenetic abnormality that identifies a MDS subset, along with morphological features [91]. Del(5q) MDS is observed in 5–10% of cases, showing severe anemia, neutropenia, and a normal or increased platelet count. These patients display a good prognosis and generally respond to lenalidomide treatment [91]. To establish the pathogenic molecular features associated with del(5q), a set of studies was carried out, sequencing genes related to MDS, such as *SF3B1, DNMT3A, TP53, TET2, CSNK1A1, ASXL1, JAK2* [15,91,92]. Many patients showed no mutation, or only one in these genes, and in general, the pattern of mutations was similar to that of other MDS subtypes, except for *TP53,* which was markedly more often mutated in MDS with isolated del(5q) [15,91,92]. Indeed, *TP53* exhibits mutations in about 20% of del(5q) and this condition is generally associated with an unfavorable outcome, an aggressive disease course and a higher risk of transformation to AML [30,93]. Various studies have also recognized a correlation between *TP53* mutations and resistance to lenalidomide in del(5q) MDS [93,94], highlighting the appearance and/or increase of *TP53*-mutant clones. Monitoring *TP53* clonal evolution could thus predict disease progression better in these patients than just the detection of *TP53* mutations at diagnosis, as also recommended in the WHO 2016 classification [1,95,96].

Mutations in the *TP53* gene are also identified in over 70% of MDS patients with CK, defined as three or more somatic chromosomal abnormalities present in a single clone [97]. Approximately 10% of MDS have a CK and these patients represent a heterogeneous group whose OS and disease course are influenced by a wide range of chromosomal abnormalities and somatic mutations [97]. Despite a greater structural genomic instability and a high frequency of *TP53* mutations, patients with CK-MDS had less somatic mutations in other MDS-associated genes, and these discrepancies were even more evident in the *TP53*-mutant subgroup, CK-MDS [97]. *TP53*-mutant CK-MDS patients also had a notably higher BM blast proportion and lower platelet counts, two factors strongly associated with an elevated prognostic risk according to the IPSS-R [97]. Indeed, *TP53*-mutant CK-MDS patients had an OS of less than half that of non-mutant CK-MDS, relapsed quickly after different kinds of treatment, and hematopoietic clones with *TP53* mutations were enriched after chemotherapy [82,96,98].

A monosomal karyotype (MK), instead, is defined as the existence of two autosomal monosomies or one monosomy with at least one additional structural aberration [99]. Its prognostic effect on MDS patients is still open to question: while several studies suggested that MK was linked to a very low OS [100,101], other studies considered that MK should not be regarded as an independent prognostic factor, because the adverse effects of MK on prognosis may be related to CK [102,103]. However, it was demonstrated that mutations of *TP53* cluster with MK and this association has a negative impact on the prognosis of MDS patients [104]. *TP53* mutations are not associated with a specific chromosomal deletion, nor significantly associated with chromosome 17 abnormality. Further analysis suggested that the number of mutation sites, co-mutation, clonal architecture, and the VAF of the *TP53* mutation in MK-MDS all had no effect on OS [99]. Lastly, although uncommon, *TP53* mutations also occur in low-grade MDS with a non-complex karyotype. Although poorly characterized in these cases, *TP53* mutations showed a lower VAF and generally, patients with a lower VAF had a better survival [105].

### 2.6. Therapy-Related MDS

The role of *TP53* in the MDS pathogenesis is particularly important in the context of therapy-related MDS (t-MDS). The 2016 WHO classification defines “therapy-related myeloid neoplasms” (t-MNs) as MDS and AML exposed to cytotoxic or radiation therapy for an unrelated malignancy or autoimmune disease [1]. t-MDS and therapy-related AML (t-AML) are classified as one entity because of their similar pathogenesis, rapid progression from t-MDS to t-AML, and their equally poor prognosis [106]. The two main classes of chemotherapy implicated in leukemogenesis are alkylating agents and topoisomerase II (TPII) inhibitors [107]. The majority of t-MNs cases are due to the alkylating agents used: mainly melphalan, cyclophosphamide, and chlorambucil [107,108]. These molecules cause direct DNA damage; in fact, alkylation of the DNA bases leads to inter- and intra-strand cross linking, abnormal base pairing, and DNA double-strand breakage [108]. TPII inhibitors, on the contrary, not only induce DNA double-strand breakage, but also interfere with DNA replication, leading to the stabilization of double-stranded breaks, more frequent DNA repair errors and crossover recombination between chromosomes [109]. Currently, t-MNs are considered the result of selection and expansion of pre-existing clonal HSC populations, which have stochastically acquired mutations increasing their fitness and survival capability [110]. The most frequent molecular aberration in t-MNs affects *TP53,* and the gene is, in fact, mutated in about one-third of these patients [50,111,112,113]. However, it was shown that the same *TP53* mutant clones found at diagnosis had been detected at low frequencies (<1%), 3–6 years before the development of t-MN and even prior to any chemotherapy treatment, supporting the conclusion that t-MN is related to the cytotoxic selection of pre-existing chemo-resistant clones preferentially expanded after treatment [114]. The same mechanism was recently demonstrated in a cohort of MM patients, years before the evolution to t-MN [115]. In these cases, the presence of pre-existing mutant HSC clones was revealed, mainly harboring *TP53* mutations, that became the dominant population at the time of t-MN evolution [115]. This mechanism could explain the high frequency of *TP53* mutations in these patients. Another genetic event linked to the high prevalence of *TP53* mutations in t-MNs is the incidence in elderly patients (over 70 years old) with cancer of CHIP. In the general population, CHIP occurs in about 10% of individuals aged 70 years or older [116]; this incidence grows to 33% in cancer patients of the same age [117]. In a recent study it was shown that *TP53* and *TET2* are the two most commonly mutated CHIP-associated genes (38%) in these patients before t-MN evolution, and the mean VAF of CHIP mutations had expanded by the time of the t-MN diagnosis [117]. Elderly patients with mutated CHIP-associated genes (such as *TP53*) therefore have an increased risk of developing t-MNs compared with those without CHIP [117]. t-MDS generally has an aggressive clinical course, often due to *TP53* mutations. Patients usually have a poor performance status and commonly show suboptimal responses to conventional chemotherapy; allo-HSCT is therefore the only curative option [113]. The outcome of transplanted t-MDS is, in fact, similar to that of transplanted de novo MDS. Importantly, despite the high prevalence of *TP53* mutations in t-MDS, they did not affect transplant outcome, even if more studies are needed, focused on the heterogeneity of *TP53*-mutated t-MDS to identify which cases could require novel emerging treatments [113].

## 3. Clinical Implications

### 3.1. Prognosis and Clinical Outcome

In order to perform the best MDS risk stratification, several prognostic models including the IPSS, WHO based Prognostic Scoring System (WPSS), and IPSS-R have been developed [118,119]. Recently, numerous groups demonstrated the importance of integrating mutational profiling into the IPSS-R for a better MDS patients risk prognostication [5,120,121,122]. The adverse effects of *TP53* alterations on MDS clinical phenotypes and outcome (more aggressive disease and poorer response to treatment) have been widely documented; the role of *TP53* mutations as predictors of poor OS in MDS patients has been uniformly described, independently of established risk factors [97,121,122,123,124,125]. On the contrary, *SF3B1* mutations identify a subgroup of MDS patients with a relatively good prognosis. The International Working Group for the Prognosis of Myelodysplastic Syndromes (IWG-PM) has recently recognized *SF3B1*-mutated MDS as a distinct nosologic entity, characterized by ring sideroblasts, ineffective erythropoiesis, and indolent clinical course [126]. Interestingly, *SF3B1*-negative MDS with ring sideroblasts have a significantly shorter survival compared to the *SF3B1*-mutated group and a significantly higher prevalence of *TP53* mutations was reported in these patients [126]. 

In the past years, the correlation between *TP53* and MDS outcome has been studied in more detail, paying special attention to the type of *TP53* alterations (mutations vs deletions), to characterize the impact of the *TP53* mutations burden and the more complex *TP5*3 allelic state. The prognostication of *TP53* mutations (mut) and deletions (del) was studied in a large cohort of 3307 patients with hematological malignancies: AML (*n* = 858), MDS (*n* = 943), ALL (*n* = 358), CLL (*n* = 1148) [14]. MDS was the only entity in which a significant negative impact on OS was shown for all the possible *TP53* alterations vs WT patients: *TP53*mut only (19 vs 65 months, *p* < 0.001), *TP53*del only (24 vs 65 months, *p* = 0.011). and *TP53*mut+del (4 vs 65 months, *p* < 0.001) [14]. Interestingly, the impact on OS of mono and bi-allelic *TP53* alterations results very different. The same circumstance was highlighted in the last study of the *TP53* allelic state implications on MDS prognostication [55] In a cohort of 3324 patients, it was shown that the two *TP53* allelic states were associated with distinct clinical presentations and outcomes. Mono-allelic *TP53* altered patients had less cytopenia and lower percentages of BM blasts compared to multi-hit patients. About 50% of patients in the mono-allelic group were classified in IPSS-R as good/very-good risk, whereas 89% of the multi-hit group were stratified as poor/very-poor risk. Moreover, the two allelic states had very different effects on the incidence of AML transformation and on OS [55]. As already discussed, the *TP53* mutations burden has also been reported to be of prognostic significance in MDS patients [56,57,58]. In patients with mono-allelic *TP53* mutations, cases with a VAF > 23% had an increased risk of death compared to WT patients (*p* < 0.001), while cases with a VAF ≤ 23% had a similar OS to WT patients. On the contrary, multi-hit patients had poor outcomes across all ranges of VAF [55]. Other authors, analyzing 261 *TP53*-mutated MDS, developed a multivariable model for OS that included the IPSS-R categories (blast score, cytogenetic score, hemoglobin score, platelet score) and *TP53* VAF [98]. In view of these data, the generic adverse role of *TP53* alterations in MDS needs to be reconsidered. The simple presence/absence of *TP53* mutation/s is not enough to define the patient’s prognosis, and more aspects must be considered. The outcomes of *TP53* altered MDS are heterogeneous and their response and prognosis may differ on the basis of the mutation burden and genomic context, even when correcting for clinical biological aspects [98].

### 3.2. Conventional Therapeutic Approaches

The wide range of therapeutic strategies available for *TP53*-mutated MDS cases was recently excellently described [7]; herein we aim just to summarize the main standard and emerging approaches in order to show how the study of *TP53* alterations in MDS patients can influence treatment decision-making and predict the response rate. Conventional therapeutic approaches include hypomethylating agents (HMAs), lenalidomide, and allo-HSCT [7]. The approved HMAs, azacytidine (AZA) and decitabine (DAC), are the standard frontline treatment option in patients with higher-risk MDS [7]. AZA and DAC are two nucleoside analogs able to incorporate into DNA (AZA also in RNA) [127]. Their predominant effect is to inhibit DNA methyltransferase, and revert hypermethylation-induced silencing of tumor suppressor and other cancer-related genes [128,129]. It is not clear whether the *TP53* mutational status can influence the response rate to HMAs in MDS patients. Some preclinical studies have demonstrated that *TP53* mutations increase the cells sensitivity to HMAs [130,131], and an improved response rate in *TP53*-mutated MDS was reported in some clinical studies [132,133]. On the contrary, no significant differences in response rate were observed between *TP5*3-mutated and WT patients in other studies [15,134,135,136]. Interestingly, HMAs induce a *TP53* mutational burden reduction (to VAF < 5%), rarely seen with other genes recurrently mutated in MDS [133,134]. This suggests the importance of *TP53* analysis not only at the disease onset but also during the follow-up, in order to perform a molecular evaluation of the treatment response. The efficacy of HMAs in combination with other therapeutic agents is also under investigation. An ongoing trial (NCT03377725) is evaluating the use of DAC in combination with arsenic trioxide (ATO). ATO is widely reported to be able to degrade and thus inhibit the oncogenic function of mutated p53 [137]; ATO, in fact, suppresses cancer cell growth by targeting mutated p53, for degradation by Pirh2-pathway [138]. Furthermore, it was shown that the combination of ATO and DAC synergistically induces the apoptosis of MDS cells, increasing the levels of reactive oxygen species and inducing the endoplasmic reticulum stress [139]. In *TP53*-mutated MDS, the potential of DAC plus ATO combination will be evaluated considering the RFS improving and the ability to thoroughly eliminate the *TP53*-mutated subclone.

In patients with an isolated 5q deletion, characterized by severe, often refractory anemia, the standard treatment option is lenalidomide, an immunomodulatory agent that can reduce transfusion requirements and reverse cytologic and cytogenetic abnormalities [140]. Lenalidomide was shown to be able to stabilize MDM2, accelerating the degradation of p53, that is overexpressed in erythroid precursors in these patients [26]. Different clinical studies have uniformly reported a correlation between *TP53* mutations and resistance to lenalidomide in del(5q) MDS [93,94], in terms of a reduced response rate, poorer OS, the appearance and/or increase of *TP53*-mutant clones, and a higher risk of AML transformation compared to WT patients [7]. For these reasons, the conventional use of lenalidomide in del(5q) MDS should be reassessed in the presence of *TP53* mutations, in favor of other therapeutic strategies such as HMAs [7].

The only curative treatment approach for MDS patients is allo-HSCT, that should be considered in all eligible patients [141]. In *TP53*-mutated MDS, its use is still debated in view of the adverse impact on the outcome of these patients [142,143,144]. *TP53* mutations were shown to be an independent risk factor for a lower OS, higher cumulative incidence of relapse, and lower event-free survival [145]. *TP53* mutations were then significantly associated with poor outcomes after transplantation for patients with de novo MDS, primarily owing to a higher prevalence of relapse [145]. Remarkably, it was recently shown that HMAs used as a bridge to allo-HSCT could reduce the adverse outcome observed in these patients [133]. Furthermore, as previously discussed, allo-HSCT is the only curative option for t-MDS cases, regardless of the *TP53* mutational status; the outcome of transplanted t-MDS is, in fact, similar to those of transplanted de novo MDS [113].

### 3.3. Emerging Therapeutic Strategies

In the era of precision medicine, several new approaches are being developed with the purpose of supplanting or supplementing the older drugs mentioned above. The variety of therapies currently under investigation can be classified in two groups, as targeted therapies and immunotherapies. Targeted therapy aims to inhibit or enhance one of the numerous molecular pathways in which p53 is involved. The most promising agent tested to date is APR-246, a methylated derivative of PRIMA-1, which induces apoptosis in human tumor cells through restoring the transcriptional transactivation function of mutant p53 [146]. APR-246 is spontaneously converted to the reactive electrophile methylene quinuclidinone (MQ) that, in the cellular environment, form adducts with thiols (cysteine residues) in mutant p53 [147]. Covalent modification of mutant p53 per se is enough to produce thermodynamic stabilization of the protein toward the WT conformation, inducing apoptosis of the tumor cell. The interaction between APR-246 and p53 was recently better described and cysteine 277 was identified as the primary binding target for MQ in p53 [148]. Cysteine 277 is essential for MQ-mediated thermostabilization of WT, R175H and R273H mutant p53. Moreover, together with cysteine 124, it is required for the functional restoration of R175H mutant p53 in living tumor cells [148]. MQ not only reactivates mutant p53, but also targets antioxidant molecules influencing the cellular redox balance [149]. In fact, it was shown that the MQ has the ability to inhibit the selenocysteine-containing enzyme TrxR1 and to bind and deplete glutathione, inducing a cellular oxidative stress. Dual targeting of mutant p53 and the redox balance induces the elimination of cancer cells [149]. Low doses of APR-246 alone, or in combination with AZA, reactivate the p53 pathway and induce an apoptosis program. The clinical effect of APR-246, as single agent, was investigated in refractory hematological malignancies, showing its ability to target p53 in vivo [150,151]. Furthermore, the synergistic effect of APR-246 and AZA was demonstrated in *TP53*-mutated MDS and AML [146]. Several ongoing trials (NCT03745716, NCT03931291, NCT03588078, NCT03072043) show the high efficacy and tolerability of this combination regimen in terms of overall response rate (ORR) and complete remission (CR) for these patients [152]. Notably, a 74% ORR and 59% CR was shown in 27 evaluable MDS patients from the phase Ib/II trial NCT03588078. Furthermore, 88% ORR and 61% CR were reported in 33 evaluable MDS patients from the phase Ib/II trial NCT03072043. Seventeen (52%) evaluable MDS patients from this clinical study discontinued treatment to pursue allo-HSCT [152].

The other agent tested for the purpose of inducing apoptosis in tumor cells is venetoclax, an inhibitor of the antiapoptotic protein Bcl-2. p53 binds to Bcl-2 via the DBD and induces mitochondrial permeabilization, with the release of apoptotic activator proteins [153,154]. In contrast to the WT protein, missense mutants of p53 are unable to form complexes with Bcl-2 in human cancer cells. Clinical studies of venetoclax in combination with HMAs in higher-risk patients with MDS and in AML are currently ongoing. However, it is not clear if this combination is also improving outcomes in *TP53*-mutated patients [155]. The latest mechanism exploited to solve the deleterious effects of mutant p53 is its inhibition/degradation. The agents tested for this purpose are HDAC inhibitors [156], statins [157], and NEDD8 inhibitors [158]. Ongoing trials are evaluating their efficacy and synergism with the older approved drugs. 

The use of immunotherapeutic agents is the other frontline approach in oncology, developed as from the 1990s and constantly evolving. One of the most widely studied immunologic mechanisms to date is the “immune checkpoints,” and the ability of molecules such as *PD-1* and *CTLA-4* to suppress T-cell-mediated killing of cancer cells [159]. An aberrant upregulation of these genes was observed in CD34+ cells and in peripheral blood mononuclear cells of MDS patients. Furthermore, in a cohort of patients treated with HMAs, cases resistant to therapy had higher relative increments in these genes expression compared with patients who achieved response [159]. In the light of these data, the use of PD-1 and CTLA-4 inhibitors, monoclonal antibodies anti-PD-1 (nivolumab) [160] and anti-CTLA-4 (ipilimumab) [161], was considered in MDS. Moreover, a *PD-1* upregulation was shown in *TP53*-mutated cases in comparison with WT patients [162]. For these reasons, several clinical studies are ongoing to evaluate the efficacy of checkpoint inhibitors, their synergism, and the effect of their combination with conventional therapies. 

## 4. Conclusions

The high phenotypic and clinical heterogeneity of MDS patients mainly has a genetic basis. The introduction of NGS in clinical practice has considerably increased our genomic knowledge of these disorders. In the set of genes recurrently mutated in MDS, alterations of *TP53* identify a subgroup of patients with peculiar biological and clinical aspects. In fact, *TP53* plays a pivotal role in several molecular pathways implicated in cellular differentiation and the induction of apoptosis. The latest findings show multiple implications of the *TP53* allelic state on genome stability, prognosis, and clinical presentation in MDS patients; therefore, *TP53* characterization must be viewed as a part of the MDS diagnostic workup. Furthermore, *TP53* dysregulation affects the response rate to treatment, and several new approaches are being developed with the aim of supplanting or supplementing the conventional therapeutic strategies. In the era of personalized medicine, the MDS diagnostic process cannot do without a complete assessment of the *TP53* mutational profile, to provide physicians with key molecular data for patient management and to identify the patients subgroup that could benefit from targeted therapy.

## Figures and Tables

**Figure 1 ijms-21-03432-f001:**
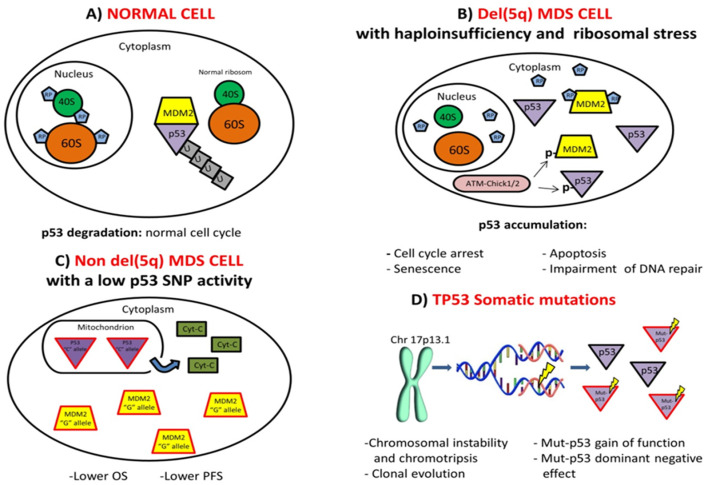
Overview of the molecular pathways activation. (**A**) A normal cell in which MDM2 is free to bind p53. MDM2-p53 binding determines the p53 ubiquitination and consequent p53 degradation, allowing a normal cell cycle. (**B**) A del(5q) myelodysplastic syndromes (MDS) cell with *RPS14* haploinsufficiency and nucleolar stress, in which ribosome assembly is impeded and small ribosomal proteins (RP) do not bind 40S and 60S ribosomal subunits but they are free in cytoplasm and bind MDM2. MDM2-RP binding avoids MDM2-p53 interaction resulting in a p53 stabilization, reinforced also from the phosphorylation of both MDM2 and p53 by ATM-Chk1 or ATM-Chk2. The accumulation of p53 leads to cell cycle arrest, impairment of DNA repair, senescence, and apoptosis. (**C**) A non del(5q) MDS cell with a low p53 SNP activity in which the *MDM2* SNP309 “G” allele enhances *MDM2* expression and p53 “C” allele has an improved apoptosis-promoting potential, in part for its major mitochondrial positioning, activating cytosolic liberation of cytochrome C. This condition determines in patients a significantly lower overall survival (OS) and progression-free survival (PFS). (**D**) Somatic *TP53* mutations are related, in MDS patients, with p53 gain of function and dominant negative effect, chromosomal instability, chromotripsis, and clonal evolution.

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
