# Peer review of "TP53 in Myelodysplastic Syndromes: Recent Biological and Clinical Findings"

_ijms, 2020, doi:10.3390/ijms21103432_

Round 1
Reviewer 1 Report
The review article by Cumbo and colleagues comprehensively describes the current updates in the clinical and therapeutic advances of TP53 mutated MDS.
The article is well written and the use of references is appropriate.
The reviewer thinks that the therapeutic part is lagging behind the clinical description and must be improved before the work is accepted for publication.
In particular:
- authors must describe the mechanism of action of APR-246, the most clinically advanced mutant p53 reactivating compound discovered by Klas Wiman and colleagues based on the most recent reports.
- authors must include the description of the ongoing trials with arsenic trioxide and decitabine
- it would be of relevance to the readers to include the newest update on the clinical outcome of APR-246 in TP53 mutated MDS trials
The new study adding to the classification of MDS must be included in the clinical description Malcovati et al SF3B1-mutant myelodysplastic syndrome as a distinct disease subtype - A Proposal of the International Working Group for the Prognosis of Myelodysplastic Syndromes (IWG-PM), Blood 2020.
Reviewer 2 Report
Cumbo et al., write an extensive and well referenced review of the role of p53 in MDS. Apart from some organizational issues which should be easily remedied the article should be published with minor changes.
Major Comments
The introduction is accurate and while written but could tie the article together better. I suggest reordering the introduction to briefly introduce MDS syndromes, followed by a discussion of p53 loss/mutation as a prominent risk factor for AML development, and ending with an introduction of p53 as a molecule.
A discussion of the potential role of clonal hematopoiesis, the prevalence of p53 mutations in that disorder, and its link to MDS would strengthen the article. This is discussed briefly towards the end, but I think an earlier and more prominent discussion would be helpful.
Most generally though the individual sections are well referenced and comprehensive the overall review is sometimes difficult to follow as it lacks overall flow and connections between sub-sections. A reorganized (see above) and slightly expanded introduction would be very helpful.
Minor comments
Line 55—missing word: CDC25C is a phosphatase that regulates cell division
Lines 135-45 -- The discussion of the role of p53 in chromosomal instability in MDS is confusing. It is not clear with the reader should take away from this experiment. It should either be discussed more or less.
Line 256-260 – I agree with the author that NGS approaches are important for accurate VAF calls. I am more skeptical on the role of long read sequencing, which needs to overcome accuracy problems before it will be of clinical utility. A note of caution would be helpful here.
Lines 305-309 – Some numbers (eg. pvalues, correlations R, etc…) would strength these points about an association and give a useful feel for the relative value of IHC p53 detection.
Lines 376-8 – Topoisomerase 2 inhibitors (eg. etoposide) do not only interfere with DNA replication, as shown by the damage they cause to cells in G1 for example. These agents principally act through inducing double strand breaks (and generating topo2-DNA lesions) both during and outside of s-phase.
Lines 489-493 – The authors should add an additional note of caution to their discussion of PRIMA derivatives. The mechanism, specificity, and viability of this strategy has not been robustly demonstrated.
Round 2
Reviewer 1 Report
The authors have now addressed most of the concerns raised by the reviewer.
Nonetheless, the thing that is confusing is:
line: 550 - 554
'Two ongoing trials (NCT03745716, NCT03931291) are showing the high efficacy and tolerability of this combination regimen in terms of complete response and deep molecular remission achievement for these patients [151]. However, the PRIMA derivatives’ mode of action, the specificity and viability of their use in 553 clinical practice need further investigations.'
The authors refer to the publication from 2018 - ref 151. There is a newer reference describing the update from the clinical trials in TP53 mutated MDS with APR246 (Liren et al., CDDis 2020) summarizing the clinical efficacy of the APR-246 compound. In addition, Klas Wiman's Lab has clearly described the known mechanism of action of APR-246 in p53 as a hub in cellular redox regulation and therapeutic target in cancer. J Mol Cell Biol. 2019 Apr 1;11(4):330-341. doi: 10.1093/jmcb/mjz005. The authors should refer to this work as well.
